# An Overview of the Copper Oxide Nanofillers Integrated in Food Packaging Systems

Kasi Gopinath [1], Gnanasekar Sathishkumar [1,*] and Liqun Xu [1,2]

[1] School of Materials and Energy, Southwest University, Chongqing 400715, China; gopiscientist@gmail.com (K.G.); xulq@swu.edu.cn (L.X.)
[2] Key Laboratory of Laser Technology and Optoelectronic Functional Materials of Hainan Province, College of Chemistry and Chemical Engineering, Hainan Normal University, Haikou 571158, China
* Correspondence: sathish88@swu.edu.cn

**Abstract:** Recently, functional nanomaterials with unique sizes, shapes, and surface chemistry have been fabricated for various applications in all facets of science and technology. Among these diverse nanomaterials, copper oxide nanoparticles (CuO NPs) have garnered considerable attention due to their unique physicochemical parameters and semiconductor properties. Doping various functional materials in CuO NPs and the fabrication of CuO nanofillers functionalized with natural or synthetic moieties delivers improved antibacterial efficacy in food packaging applications. Moreover, the bactericidal effect of modified CuO NPs against foodborne pathogens largely contributes to their usage in food packaging technology. Therefore, it is essential to fabricate effective antimicrobial CuO nanofillers with minimal or no adverse side effects. This review discusses the synthesis, characterization, surface modification, antibacterial properties, food packaging applications, and toxicological implications of the diverse CuO nanofillers integrated in films and composites. In addition, it highlights their adverse side effects and ways to combat adverse situations. The forthcoming generation is expected to lead a groundbreaking surge of inventive food packaging systems (FPS) based on CuO hybrid nanofillers in food packaging industries.

**Keywords:** copper oxide; nanofillers; synthesis; antibacterial; food packaging; toxicity

## 1. Introduction

Due to the increasing need for ready-to-eat fresh food products worldwide, it is important to produce a variety of eco-friendly and safer food packaging systems (FPS) for preserving/storing food products and preventing spoilage [1,2]. Most currently employed natural and synthetic polymeric FPS have shown drawbacks such as non-degradability, poor physical barriers, and weak mechanical properties. In particular, mechanical damage shortens food products' shelf lives because of colonization by foodborne contaminants (bacteria, fungus, and viruses), which results in economic, health, and environmental risks [3–5]. According to the Centers for Disease Control and Prevention (CDC), around 48 million individuals in the United States suffer from foodborne diseases each year, resulting in nearly 3000 fatalities [6]. Recent reports have stated that the increasing demand for FPS has led to the establishment of a worldwide market worth more than USD 300 billion and with a 5.2% annual growth rate. Among these FPS, plastic packaging materials dominate the food packaging industry, with 61.2% of the global market share, which is alarming in the context of future environmental and pollution concerns [7]. Therefore, it is imperative to develop nanofiller-based active FPS with improved physically protective properties to attain satisfactory shelf lives, food quality, and food safety. Nanopackaging should be feasible for dipping, volatile exposition, antibacterial protection, low-temperature storage, modified atmosphere packing, and edible coatings. Many polymers, inorganic nanomaterials, and biomacromolecules have largely demonstrated the potential for use in active and smart food packaging applications [8].

Using inorganic nanofillers in FPS has drawn the attention of researchers due to their stupendous mechanical qualities and efficiency in inhibiting/eradicating microbial contamination to expand food products' shelf lives [9,10]. Accordingly, nanocopper oxide (CuO) is a multifunctional transition metal oxide with unique features, such as its highly specific surface area, electrical conductivity, suitable redox potential, and excellent stability [11]. CuO nanoparticles (NPs) are well known for their easy availability, low toxicity, and ease of synthesis by various means. The above properties make nanocopper oxide a suitable agent in multiple areas, including electrochemistry, catalysis, sensors/biosensors, and biomedicine. In medical and food packaging technology, CuO nanocomposites have been effectively used as antifouling, antimicrobial, antioxidant, anticancer, catalytic, drug carrier, and biosensing agents [12–14].

In addition, modifying the surfaces of CuO NPs with other agents like nanomaterials, polymers, and bioentities further augments their properties and efficiency. Considering this, CuO-based nanocomposites have emerged as viable nanofillers and coatings to avoid the deterioration of eatables [15–18]. CuO nanofillers are incorporated into a polymer matrix with strong coordination and the prolonged release of bactericidal chemicals to eradicate contaminants [19–22]. It is now well established that the bacteria-killing effect of CuO NPs profoundly depends on their size, shape, surface chemistry, and dosage [23–25]. More precisely, CuO NPs with a high surface area/volume ratio trigger bacterial cell membrane disruption and leakage due to excessive reactive oxygen species (ROS) generation. Other phenomena suggest that the electrostatic interaction of CuO NPs with negatively charged bacterial cell membranes results in the intracellular release of $Cu^{2+}$ ions. Subsequently, the $Cu^{2+}$ ions interact with sulfur-containing biomolecules, impairing their functions [26,27]. Also, the complexity of bacterial cell membranes plays a crucial role in Cu-based products' antibacterial mechanisms and efficacy levels [28]. However, there are quite a few drawbacks (cost, impurity, toxicity, solubility, stability, and biocompatibility) related to the manufacture and uncontrolled use of CuO nanoproducts that must be examined and addressed [29,30].

The increasing exploitation of CuO NPs in food nanotechnology necessitates caution and regulations on their possible toxicological risks to protect human health and the environment [31]. Inorganic nanofillers exploited for food packaging applications could migrate from FPS into food, posing a toxicity risk once ingested. It is essential to perform validation experiments on their percentage of migration, eco-toxicological aspects, and biocompatibility to translate developed CuO-coated FPS into industrial/commercial food packaging applications. Moreover, adequate promotional activities to raise customers' understanding and awareness of nanopackaging will streamline this innovative technology's adoption as a sustainable approach [7].

This review summarizes the preparation of the various CuO nanofillers integrated/coated in polymeric FPS with improved mechanical and antimicrobial effects. It deliberates the formulations of Cu with diverse polymers, nanomaterials, and biomacromolecules and their coordination to attain improved mechanical properties. The antimicrobial potential of using CuO nanofillers to act against foodborne pathogens and their mechanical aspects are described in detail via constructive discussions. Last but not least, the potential toxicological implications of CuO-based FPS are highlighted to discuss their successful translation into the market and future research prospects.

## 2. The Preparation of the CuO Nanofillers Incorporated in Hybrid Composites/Films

### 2.1. The Synthesis of Antibacterial CuO Nanofillers

Copper forms two types of oxides—cuprous oxide ($Cu_2O$), which has a yellowish brown color, and cupric oxide (CuO), which has a black color—with small band gap energies and high stability. Generally, CuO NPs and composites can be synthesized in either top-down or bottom-up strategies. A myriad of techniques, such as laser ablation [32,33], the hydrothermal method [34], microwave irradiation [32], sonication [35], and chemical and biogenic reduction [32,36–38], have been developed to fabricate antibacterial CuO NPs and their composites. The hydrothermal method is a precise technique for synthesizing

CuO nanomaterials due to its facile, cost-effective process and high-yield end products. For hydrothermal synthesis, the metal precursors and surfactant mixture are exposed to different temperatures and significant vapor pressure. The morphology and growth of CuO nanostructures can be controlled thermodynamically in a wet chemical process [39]. George et al. [40] unveiled that calcination temperature plays a crucial role in CuO size, crystalline nature, bandgap, and surface properties. Upon exposing the reaction mixture at very high calcination temperatures (800 °C), the crystallites break due to the thermal agitation of atoms and generate NPs with reduced crystallite size. Interestingly, the monodispersity and increased surface-to-volume ratio of CuO NPs led to remarkable antibacterial effects against Gram-positive and Gram-negative pathogens in [24,41].

According to a previous report, the size and shape of CuO nanostructures can be controlled using stabilizers/capping agents in the hydrothermal method. The surface characteristics of CuO NPs are tunable according to the capping agent and pH of the reaction medium [25]. In the last decade, several studies have been undertaken to synthesize CuO NPs, prompting the exploitation of diverse plant and microbial entities as potential stabilizers. Compared with toxic chemical stabilizers, the bio-fabrication of CuO NPs has received enormous interest due to their sustainability and green chemistry principles for medical and industrial applications [42–46]. A plentiful amount of active biomolecules, such as polyphenols, proteins, carbohydrate polymers, and extracellular constituents, have also been used as stabilizers in CuO nanocomposite preparation. Gvozdenko et al. demonstrated the direct precipitation method to fabricate gelatin-functionalized CuO NPs. The copper precursor and gelatin mixture was kept at 90 °C heat under constant stirring with the addition of an alkaline solution, resulting in CuO precipitates. In an aqueous medium, highly monodispersed gelatin-stabilized CuO particles with a mean diameter of $18 \pm 6$ nm were formed [15].

As shown in Figure 1a, CuO NPs synthesized by various means appear as a brownish black powders with a monoclinic structure of crystalline axes, namely a, b, and c, with specific orientations [47–49]. As-prepared chitosan (CS)-based CuO NPs prepared using *Trianthema portulacastrum* leaf extract exhibit an absorption peak at 295 nm, mainly due to the surface plasmon resonance (SPR) effect. It has been revealed that the size, shape, surface chemistry, refractive index, and inter-particle distance of CuO NPs generate a distinctive SPR effect [50]. Cherian et al. [51] noticed that the –OH, –NH(C)=O, and –COOH functional groups of biopolymer sodium hyaluronate contribute dipole–dipole interactions and H-bonding to form stable CuO NPs (Figure 1b). Scanning electron micrographs of CuO NPs fabricated using different temperatures (75, 100, and 150) display elongated particles and platelet structures [39].

Metal-doped CuO nanofillers were prepared using a co-precipitation technique by mixing metal precursors and a surfactant in an alkaline medium in [52]. Gopinath et al. recently fabricated Neodymium (Nd)–Cadmium (Cd)-doped CuO nanocomposites in a different stoichiometric ratio. A significant imbalance between the Nd and Cd ions in the CuO lattices was seen in the Cu/Nd/Cd (94:5:1) and (94:4:2) samples, resulting in some noticeable rod-like formations. Interestingly, ensuring that the Cu, Nd, and Cd are in a 94:3:3 ratio (Cu/Nd/Cd) facilitates the growth of long rod-like structures with sizes ranging between 26.24 and 211.60 nm. The equal molar ratio of Nd and Cd promotes the growth direction of the 2D nanorods over 200 nm [53].

Integrating active functional dopants into the crystalline structure of an intrinsic semiconductor (pure) forms the extrinsic semiconductor with modulated structural, optical, catalytic, and electrical properties [54,55]. The dopant can be an electron donor or acceptor, which acts by changing the energy state or shifting the energy bands after integrating with a crystalline structure. Usually, dopants with minute atomic numbers can alter their electrical conductivity upon addition during or after crystal growth. Doping plays a crucial role in modulating the different properties of NPs due to the following reasons: (1) the modification of the electronic structure and surface state, (2) the incorporation of new functional groups, (3) enhanced catalytic properties, (4) improved photoexcited hot carrier

generation, (5) greater stability, and (6) better biocompatibility [56]. As shown in Figure 2, doping agents can be metallic (common metals and metal oxides) or non-metallic agents. A few other dopants, such as polymers, macromolecules, sulfur, nitrogen, chloride, and nitrates, have been used to enhance the properties of CuO NPs [57–61]. Most of the reported studies suggest that the concentration of metal precursors, stoichiometry, temperature, pH, and reducing and stabilizing agents play a crucial role in the structural and functional features of CuO nanofillers. Developing optimized, simple, and reproducible protocols to fabricate antibacterial CuO nanofillers with minimal downstream processing for scaled-up processes is always recommended.

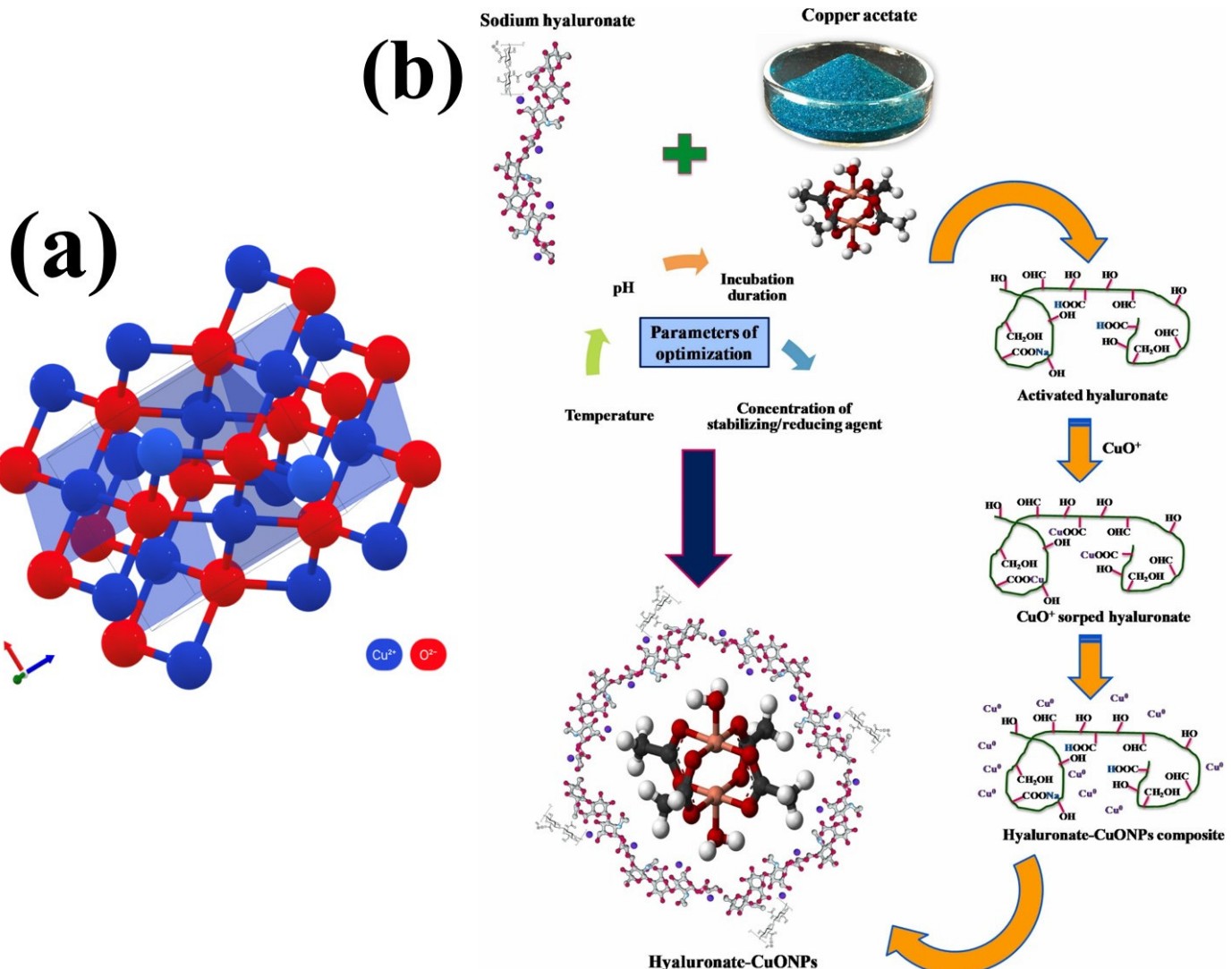

**Figure 1.** (**a**) The monoclinic crystal structure of CuO. (**b**) The mechanism of hyaluronate-assisted CuO NPs synthesis. Reprinted with permission from [51] Copyright © 2023, Elsevier.

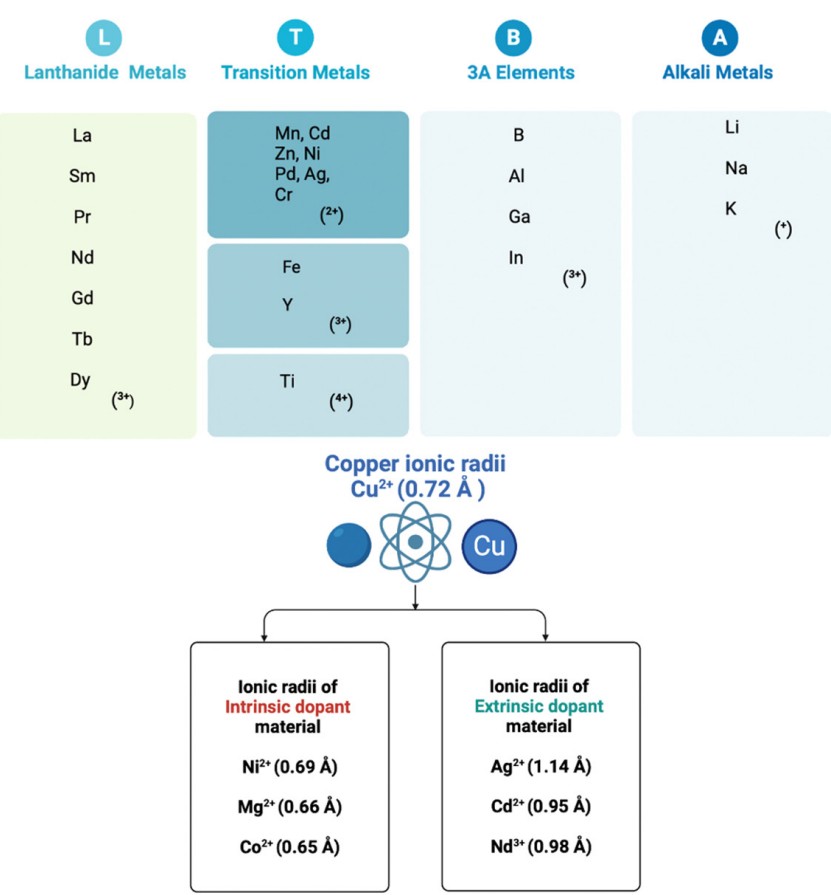

**Figure 2.** The different oxidation states of the metals incorporated in CuO lattices and the different ionic radii of the metals incorporated in CuO lattices by intrinsic and extrinsic dopants. This diagram was created using BioRender.com (accessed on 25 August 2021).

### 2.2. The Effects of CuO Nanofiller Integration in Polymeric Matrices

In nanocomposites, one or more nanosized particles are integrated into a matrix of standard material for tailor-made applications [62]. It has been exemplified that natural structures can achieve mechanical properties such as having strength; dimensional stability; electrical conductivity; an increased surface area and binding capacity; thermal stability; chemical resistance; optical clarity; and reduced gas, water, and hydrocarbon permeability [63–65].

Considering the above, CuO nanofillers integrated with natural or synthetic polymeric matrices have gained much attention and provide better applications. Araujo et al. [34] prepared CuO-deposited bacterial cellulose (BC) through a hydrothermal process by placing the BC membrane in a $Cu(NO_3)_2$ solution and subsequently adding $NH_4OH$ under hydrothermal conditions. The hydroxyl functional groups of BC contribute to the reduction and growth of CuO nanostructures onto the hydroxyl sites of BC fibers at 150 °C due to the alkaline hydrolysis process (Figure 3a,b). In another study, it was explicitly proved that the layer-by-layer addition of 1% CuO NPs into CS and sodium alginate (SA) composite films showed the best tensile strength (TS), lowest roughness, and smoothest surface (Figure 3c,d) [66]. Moreover, increasing the alkali ion exchange time triggers more CuO nanofiller diffusion and interactions on the surface of the polymer matrix [67].

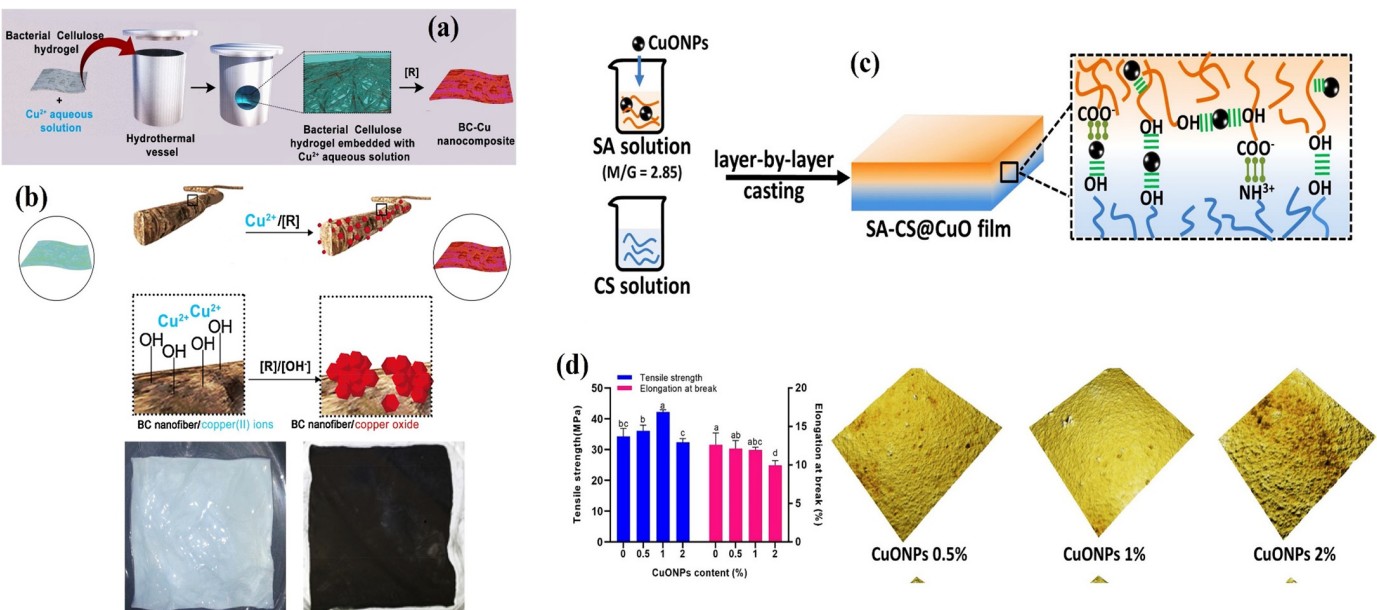

**Figure 3.** (**a**) A schematic of Cu deposition into a BC membrane surface via the hydrothermal synthesis of a BC-Cu nanocomposite. (**b**) A BC hydrogel immersed in Cu(II) aqueous solution, which after the reduction and deposition of Cu-based NPs, showed a blackish color membrane. Reprinted with permission from [34] Copyright © 2018, Elsevier. (**c**) A schematic of SA-CS@CuO film preparation. (**d**) The mechanical properties (Columns marked by the same letter are not significantly different, $p < 0.05$) and atomic force microscopic topography of SA-CS@CuO films. Reprinted with permission from [66] Copyright © 2021, Elsevier.

A CuO-reinforced biohybrid film made of polyvinyl alcohol, polyethylene glycol, and citric acid showed improved barrier properties. Also, the water solubility of this nanofilm ranges from 3.64 to 100%, depending upon the extent of crosslinking [68]. Roufegarinejad, 2022, constructed a dense and compact CuO and titanium-di-oxide (TiO$_2$) NP nanocomposite film using soy protein isolate (as the matrix). The fabricated nanocomposite possesses low water permeability, solubility, and moisture content with improved opacity and antioxidant activities, meaning it has promise for food packaging applications [69]. The incorporation of titanium nanotube (TNT)–CuO significantly ($p < 0.05$) increases the thickness and TS of carrageenan (Car) films. Meanwhile, Car/TNT−CuO films exhibited only slight improvements in flexibility and stiffness, mainly due to the high-level aggregation of nanofillers [52]. Thus, some extensive research investigations are going on around the world to evaluate the use of CuO-based nanocomposites in modern food packaging technology.

## 3. CuO Nanofiller-Based Antibacterial FPS

With the increasing demand for high-performance food packaging materials, NPs of various elements have gained much attention worldwide. As mentioned earlier, the engineering of the various functional nanofillers integrated in FPS bestows high thermal, mechanical, and antibacterial properties [70,71]. Moreover, the shelf lives and quality of food products can also be enhanced, in addition to their storage, processing, and transportation [72] (Figure 4).

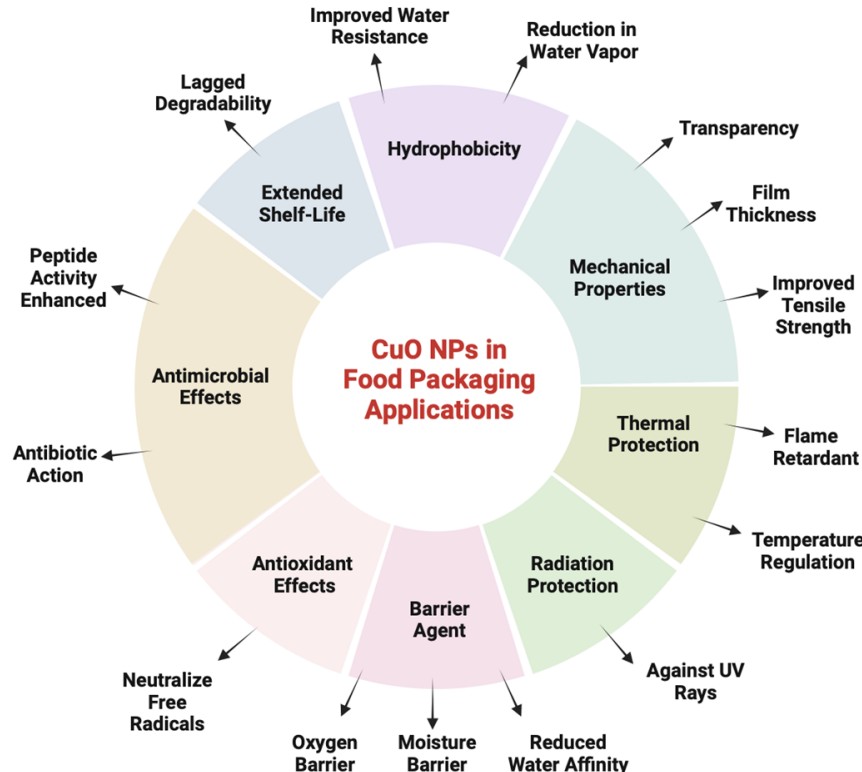

**Figure 4.** The properties of CuO nanofillers and their applications in food packaging. This diagram was created using BioRender.com (accessed on 25 August 2021).

Recently, CuO NPs have been extensively used in food packaging research, imparting several benefits that ultimately aid economic development. Cu acts as an electron donor and acceptor, making it a potential agent to inhibit a broad spectrum of food-spoiling microbial contaminants. The qualities of packaging materials have been further enhanced by doping CuO NPs with other active nanomaterials that work at the crystalline level [73–75].

*3.1. The CuO NPs Integrated in FPS*

Several studies have documented that including CuO NPs in polymeric films enables the cost-effective production of stable, eco-friendly, and viable FPS. In particular, the CuO NPs integrated in food packaging materials bestow outstanding UV blocking, hydrophobic, electrical, and thermal stability. The integration of CuO NPs increases the thickness and moisture of the polymeric films in a concentration-dependent manner. Further, antimicrobial experiments have unveiled that adding copper nanowires (CNW, 0.5%) into a SA (3%)-CuO NP (5mM) film led to the film displaying notable antimicrobial activity. A higher zone of inhibition was observed in the following: *S. aureus* > *Salmonella* sp. > *C. albicans* > *E. coli* > *Trichoderma* spp. In addition, the CNW (0.5%)-SA (3%)-CuO NP (5mM) film demonstrated potent DPPH and ABTS scavenging activities via transferring its electron density to the free radicals. A fresh-cut pepper (FCP) coated with an as-prepared CNW-SA-CuO NP film inhibited microbial contaminants more effectively than a pristine FCP, which indicates that the coated FCP would have a better shelf life, even if encased in conventional food packaging [17]. Based on these reported studies, it is proposed that the bactericidal effect of CuO occurs in three different mechanisms (Figure 5).

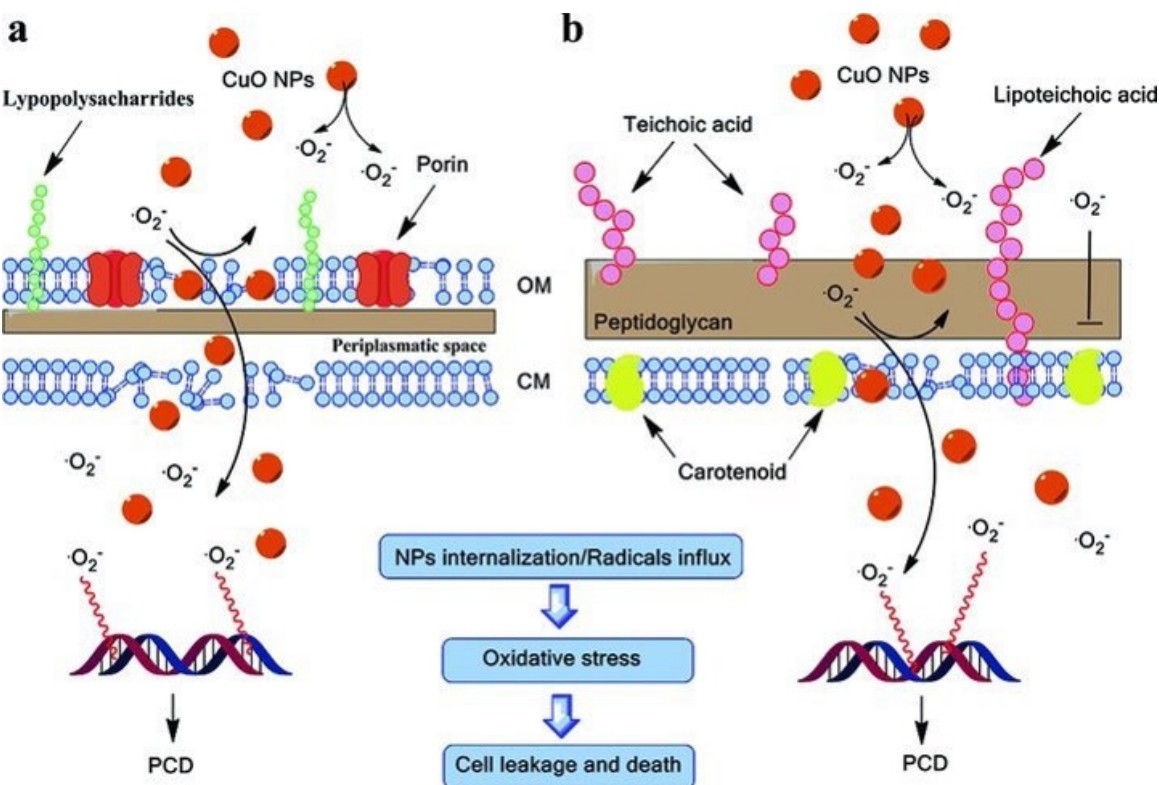

**Figure 5.** A schematic illustration of the antibacterial mechanism of CuO NPs and their relative cellular structures (**a**) *E. coli* (Gram-negative) and (**b**) *S. aureus* (Gram-positive). Reprinted with permission from [76] Copyright © 2012, John Wiley and Sons.

1.  $Cu^{2+}$ released from the polymeric films interacts with sulfhydryl groups of bacterial cell membranes and triggers enzyme inactivation.
2.  ROS generated through Fenton-type redox reactions and the electron transfer from $Cu^{2+}$ leads to more free radical-mediated oxidative bacterial killing.
3.  The sustained release and accumulation of CuO nanoparticles onto the bacterial membrane surface alter its permeability and the successive leakage of intracellular contents.

Duffy et al. [77] demonstrated that CuO NPs exhibit significant antimicrobial effects against the foodborne pathogens *Salmonella enterica*, *Listeria monocytogenes*, and murine norovirus. However, CuO NPs require higher element concentrations than Ag- and ZnO-based NPs, mainly due to the Cu resistance mechanisms of *Salmonella*. In contrast, integrating CuO NPs into biodegradable poly(3-hydroxybutyrate-co-3-hydroxyvalerate) (PHBV) improves water vapor permeability and oxygen barrier properties [20]. The high stability of CuO nanoparticles is an essential criterion for their uniform size distribution onto composite films and for limiting the migration of Cu into food products. It was noticed that methylcellulose films integrated with gelatin-stabilized CuO NPs exhibit only 0.12 μg/mg of Cu migration, which can be further reduced by keeping the food products in a refrigerator at 0–4 °C conditions. Interestingly, the developed nanopackaging delayed the spoilage of strawberries and tomatoes on the 4th and 7th days by inhibiting the growth of various microbial contaminants [15]. NanoCuO-treated low-density polyethylene (LDPE) reduced 4.21 log CFU/g of coliforms and protected cheese from microbial contaminants after one month of refrigerated storage [78]. It was clear that the integration of CuO nanofillers led to the exhibition of a broad spectrum of antimicrobial effects with improved water repellency and mechanical behavior.

### 3.2. Metal/Metal Oxide-Doped CuO NPs in FPS

Combining metal and metal oxides with CuO NPs has imparted remarkable mechanical antimicrobial, antioxidant, and UV protection properties for food packaging applications [79–81]. Doping metal oxides onto CuO NPs is a novel approach to achieve better applications owing to their modified structural and functional parameters [82,83].

Nanometal oxides usually act as fillers in polymeric matrices of FPS to improve their inherent disadvantages. As a food contact material (FCM) approved by the USFDA, $TiO_2$ NPs are often utilized as polymer fillers due to their easy preparation, affordability, non-toxicity, and stability. Wang et al. [52] demonstrated that integrating $TiO_2$/CuO NPs into a poly(butylene succinate-co-terephthalate)-based film (PTC) upgraded its water vapor permeability (WVP), UV protection ability, and antibacterial properties. The WVP values of PTC films increased with the integration of 5% $TiO_2$/CuO composites, which signifies that the decreased WVP at a lower $TiO_2$/CuO content was mainly due to crystallization hindrance. In other words, the increased crystallinity forms more impermeable regions in the matrix and makes the path tortuous. The films could absorb UV-B, UV-A, and visible light upon increasing the content of $TiO_2$/CuO NPs and protect food products' nutritional values. Among the groups tested for preserving cherry tomatoes, the PTC-3 film (3% $TiO_2$/CuO NPs) possessed the best preservation effects mainly because of its lower WVP, the while PTC-9 film (9% $TiO_2$/CuO NPs) exhibited a high-level bactericidal effect.

The high-level photobactericidal effect of a CuO-doped TNT-incorporated film, apparent upon visible light exposure, can be attributed to its significant band gap energy. It was unveiled that the TNT-Cu-generated superoxide ions ($^\bullet O^{2-}$) and hydroxyl radicals ($^\bullet OH^-$) actively inhibit bacterial DNA replication and induce protein damage and cell lysis. Compared with other groups, bananas stored in Car/TNT−CuO films retained 40%−60% of their initial firmness and edible quality (Figure 6a) [84]. In a later study, a $Cu_2O$-modified TNT composite coating was modified for cellulose nanofiber (CNF)-based film fabrication for active packaging applications in the context of fruits and vegetables. It was also validated that tomatoes packaged in CNF/TNT-$Cu_2O$ films started to change color, becoming slightly orange after 14 days of storage. The tomatoes stored using other films (CNF/$TiO_2$ and CNF/TNT) became dark and light red (Figure 6b). These results proved that Cu-doped TNT essentially reduces ethylene gas production and improves the shelf life of fruits by delaying fruit ripening [85].

Recently, a Cu-doped nano$TiO_2$ composite was exploited to prepare natural low-acyl gellan gum (LAGG)–BC composite films with improved UV protection, antioxidant activity, and flexibility. Among the different LAGGs designed using various CuO contents, the LAGG-BC/T-0.4%CuO group better maintained the quality of fresh-cut peppers. The mechanism for extending food products' shelf lives includes the LAGG-BC/T-0.4%CuO film's high barrier properties which prevent oxidative damage and browning and its ability to slow nutrient consumption and softening from water loss. Moreover, the LAGG-BC/T-0.4%CuO group had the highest hardness but the lowest weight and rot rates (<5%) throughout storage. After 6 days, these measures decreased by 11.27% and 26.07%, significantly less than the PE group's decreases of 23.64% and 52.13%. Additionally, the weight loss and rot rates of the LAGG-BC/T-0.4%CuO group were much lower than those of the control group. As a result, the LAGG-BC/T-0.4% CuO film was highly effective in preventing the reddening, browning, softening, and rotting of fresh-cut peppers, demonstrating prominent preservation effects (Figure 6c,d) [16]. Likewise, the addition of Cu-based ternary nanocomposite (Ag+Cu/$TiO_2$) to form LDPE films strongly inhibits the growth of *E. coli* and *L. monocytogenes* in seafood (Nile tilapia) packaging [86]. These reported studies showed that the synergistic effect of two metal oxides contributes to improvements in antibacterial effects and food storage.

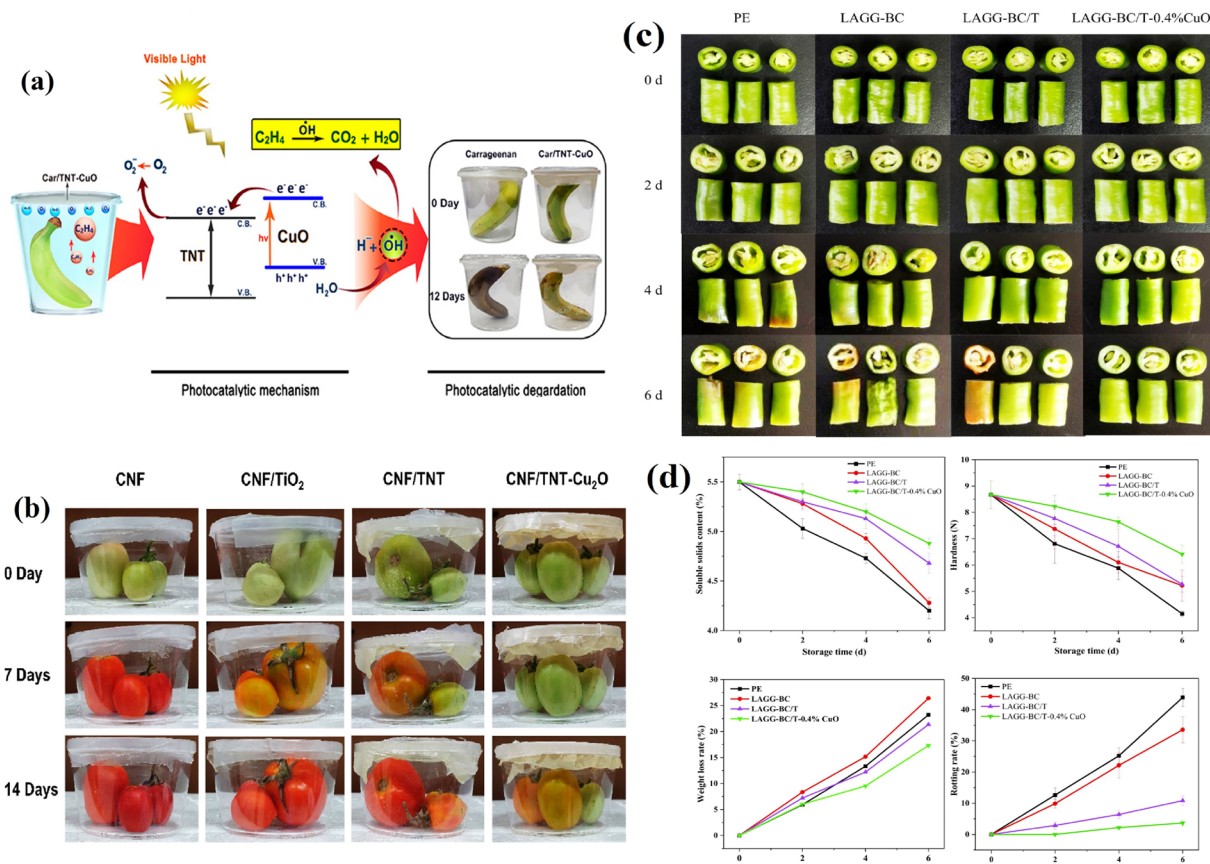

**Figure 6.** (**a**) Scheme representing how a Car/TNT−CuO film boosted the shelf lives of packaged bananas under visible light. Reprinted with permission from [84] Copyright © 2021, American Chemical Society. (**b**) Apparent changes in the color of tomatoes covered with CNF-based films during storage at 25 °C. Reprinted with permission from [85] Copyright © 2022, American Chemical Society. (**c**) Appearance and (**d**) changes in color over time of fresh-cut peppers packaged with the LAGG-BC-based films during storage at 25 °C. Reprinted with permission from [16], Copyright © 2023, Elsevier.

### 3.3. The Co-Integration of CuO–Essential Oil in FPS

Adding natural food preservatives and Cu nanofillers is considered a safe approach to prolonging the shelf lives of meat products due to their excellent functional properties. Hasheminya et al. explored using Satureja khuzestanica essential oil (SEO) and CuO NPs at different concentrations (0, 1, 1.5, and 2%) for the preparation of kefiran–carboxymethyl cellulose nanobiocomposite films. The co-incorporation of 2% CuO NPs–2% SEO lowered the moisture content of the films due to the reduced amount of hydroxyl groups for water absorption and the reduced formation of hydrogen bonds between the CuO NPs and SEO. It was noticed that the films containing different concentrations of CuO NPs exhibited dose-dependent bactericidal effects against *S. aureus* and *E. coli*. Also, the high antibacterial effect recorded against *E. coli* compared to *S. aureus* indicates that the mechanism of action of CuO largely differs depending on cell wall complexity and $Cu^{2+}$ ion interactions [87].

Among the diverse bioactive essential oils, the loading of clove essential oil Pickering emulsion (PEC) in gelatin/agar-based multifunctional films containing ZnO@Cu has shown increased free radical scavenging effects due to the presence of high-content polyphenols. As compared with the control (22 meq/kg), the reduced peroxide value (16 meq/kg) of the pork loin meat stored in the ZnO@Cu/PEC film indicates that glove oil plays a prominent role in preventing lipid oxidation [88]. Similarly, adding cinnamon essential oil (CEO) and bimetallic nanoparticles (Ag-Cu) to LDPE sheets showed significant effectiveness for chicken meat packaging. It was unveiled that the addition of CEO exhibits a robust

bactericidal effect against *S. typhimurium*, *L. monocytogenes*, and *C. jejuni* via affecting cell wall structure enzyme activity, inducing protein denaturation, and cell death. The substantial reduction in the bacterial contaminants in the chicken samples stored using the LDPE composite film (even after 21 days of storage) indicates its potential for packaging applications [89]. Hence, biocomposites based on combining CuO and essential oil can be greener approach in fresh meat packaging applications.

## 4. Toxicological Risk Assessment

The toxicological implications of using CuO NPs on human health and the environment depend chiefly on their physicochemical properties, such as their dissolution, aggregation, stability, reactions, and magnification. For example, high-dose nanocopper particles can cause liver and brain damage in rats and could transfer to other organs and tissues and further induce specific lesions [29,90]. Considering the consequences of the excessive use of CuO NPs, it is necessary to pay more attention to understanding their adverse impacts, which are mainly due to their migration into food products, and finding regulatory measures to control human exposure to these nanoparticles.

### 4.1. The Migration of CuO NPs into Food Products

Low concentrations of inorganic nanofillers have commonly been used to improve conventional food packaging materials' mechanical and antimicrobial functionalities [91]. However, the migration of NPs or ions and the potential of them coming into contact with food/edible items puts consumers at serious risk of adverse health effects. Considering this, it is reasonable to validate the migration of CuO nanopackaging systems before making them available on the market. It is now widely understood that the migration and hazardous effects of nanoparticles are heavily influenced by their size, shape, and surface behavior [31]. The migration of CuO NPs into food products presents complex challenges, including health concerns and impacts on food quality and regulatory compliance. However, simulation models are valuable tools for assessing CuO NP migration in food products due to their cost-effective, predictive, and efficient nature, aiding our understanding and management of the potential risks associated with nanomaterial migration and thereby contributing to food safety and regulatory compliance.

In response to concerns regarding NP migration, Cushen et al. [29] investigated the migration of nanosilver and nanocopper from polyethylene (PE) nanocomposites to chickens. Under various storage conditions, silver's migration levels ranged from 0.003 to 0.005 mg/dm$^2$, while copper's range was 0.024 to 0.049 mg/dm$^2$. Notably, neither time nor temperature had a significant impact on migration. Based on the Williams−Landel−Ferry equation, the migration model accurately predicted nanosilver migration but exhibited less accuracy for nanoCu, likely due to variable Cu levels in the food matrix.

Jiang et al. [92] studied the incorporation of nanoCu into a homopolymer polypropylene (PP-H) matrix to produce a composite film apt for food packaging. The migration test revealed that when the initial concentration of nanoCu added to PP-H reaches 1% (*w/w*) and the actual concentration within the film stands at 0.79%, Cu tends to migrate from the film to a 3.0% acetic food simulant, reaching concentrations of 4.5 µg/g at 70 °C. This migration has potentially harmful effects on human L-02 liver cells and the results provided crucial insights into the feasibility of using nanocopper/PP-H composite films in food packaging. The migration levels of Ag and Cu from a FCM surface coating in 3% *v/v* acetic acid exceeded the levels outlined in European Commission Regulation No. 10/2011 (European Commission (EU) 2011). However, the NPs and block copolymer combined coating showed less migration and an acceptable margin of exposure, possibly due to the relative sizes of Cu-based NPs affecting dissolution rates [93]. Shi et al. [94] investigated factors like temperature, exposure time, polypropylene (PP) types, coupling agent (KH550), and sterilizing conditions that influence copper migration from nanocopper/PP composite films into food and food simulants. It was observed that copper migrated the most from the PPH films (up to 34.51%), especially into 3% acetic acid. Gamma irradiation notably

increased Cu migration, whereas the addition of KH550 had minimal effects. Cu migration varied based on the food simulant's acidity, with rice vinegar showing the highest migration at 0.65 mg/kg. Inductively coupled plasma mass spectrometry (ICP-MS) detected copper nanoparticles in 10% EtOH but not in 3% HAc since the Cu dissolved in the acidic environment. In an acidic food simulant, Cu readily converted to ionic form because of the high hydrogen ion concentration. A comparison involving rice vinegar, apple vinegar, and lemon soft drinks revealed that the migration of Cu-based NPs in acidic foods is correlated with their pH levels.

### 4.2. Degradation and Ecological Impact

The levels of environmental toxicity induced by CuO NPs depend chiefly on their physicochemical properties, such as their dissolution, aggregation, stability, reactions, and magnification. Apart from terrestrial organisms, their presence in aquatic systems has led to the generation of sensitivity and toxicity to various species, such as rotifers, algae, crustaceans, fishes, amphibians, and others, leading to a decline in their population [95,96]. In plants, CuO NPs are known to cause toxicity by inhibiting the germination of seeds, decreasing root and shoot length, reducing respiration and photosynthesis, and changing morphology [97]. Considering the environmental consequences of nanopackaging trash, more attention should be paid to understanding its degradation and toxicity mechanism, reducing its use, and finding biodegradable alternatives. The biodegradation and recycling processes of CuO nanofiller-integrated FPS are in their infant stages. To minimize environmental risks, more in-depth experimental studies on integrating CuO nanofillers in natural polymeric FPS and their degradable mechanism should be conducted [2,83].

### 4.3. Biocompatibility Test

The use of CuO NPs is exponentially rising in industrial and commercial products for innumerable applications. Fathi et al. [18] fabricated carboxymethyl chitosan (CMCS)-composite films integrated with saffron petal anthocyanin (SPA)-CuO and tested their cytotoxicity on human dermal fibroblast (HDF) cells using the MTT (3-(4,5-dimethylthiazol-2-yl)-2,5-diphenyltetrazolium bromide, a tetrazole) assay. The CMCS film showed similar HDF cell viability to the control group (approximately 95% cell viability), indicating no cytotoxic effects. However, adding CuO NPs decreased cell viability (around 70%), indicating toxicity due to Cu ion release from the composite film's surface. When SPA was added to the CMCS film (the CMCS-SPA-2-CuO-1 film), cell viability increased to over 80%, possibly due to enhanced cell attachment through increased hydrophilicity. SPA might act as an antioxidant, reducing toxicity by chelating Cu ions. These results indicate that adding SPA to bio-based films improves cell adhesion and enhances the bio-safety of CMCS composite films.

In *in vitro* models of human skin epithelial cells and human embryonic kidney cell lines, CuO nanomaterials have been used for biocompatibility assessments. Typically, >90%–80% cell viability is considered non-toxic and acceptable for biological application [53,98,99]. When doped with metal, the CuO sample exhibited higher cytotoxicity than the pristine CuO nanomaterials, a difference attributed to size, shape, stability, and ion-releasing ability variations. Recently, engineered CuO with various ratios of Nd- and Cd-doped CuO nanocomposites showed biocompatibility on L132 human lung epithelial cells at 25 to 200 μg/mL. The pristine CuO nanomaterials demonstrated 80% cell viability at a 150 μg/mL concentration. Except for the Cu:Cd (94:6) sample, all the other samples maintained cell survivability between 90 and 80% at concentrations ranging from 25 to 75 μg/mL [53]. Nonetheless, the excessive production and usage of nanomaterials has made living organisms (humans/animals (higher to lower)/plants/microbes), as well as the environment, vulnerable to toxicity [100]. The key factors that make CuO NPs a toxic agent include their small size, exposure time, dose, bioaccumulation, and lack of degradation. It is evident from the studies reported in the literature that upon binding, they interact with living cells, resulting in a change in their surface chemistry. Thus, to elucidate

the toxicity of CuO NPs, it is required to understand their routes of exposure, mechanisms of interaction, and the pathways they follow to induce toxicity [29].

## 5. Conclusions and Future Outlook

The past few decades have witnessed tremendous advancements in nanoparticle fabrication and biomedical applications. No doubt, it is the need of the hour to find new ways to tackle dreadful diseases by generating nanomedicine and nanodiagnostic tools. Further, with the increase in the population, it is necessary to find ways to increase food production, preserve food products, and fortify the quality of food products for better nutritional gains. Nanotechnology is also spreading its branches to upthrow the food technology sector. While, in recent times, most of the research in the literature has focused on metallic NPs, the research on metal oxide NPs is still naive. CuO NPs have attracted attention due to their distinct physicochemical properties, ease of availability, ease of fabrication, and ease of modification to achieve desirable properties. The doping of CuO NPs modulates their properties by enhancing their structures, surface areas, stability, binding capacities, catalytic properties, and bioavailability by forming new functional moieties.

Thus, several methods and agents have been used to improve CuO NPs by doping polymers and other metal oxides. Combining CuO-based nanofillers in polymeric films and composites has demonstrated enhanced physical, mechanical, and bio functionalities. The potency of CuO has been amplified significantly with the fabrication of CuO nanocomposites due to the blending of different materials. However, a better understanding of the mechanisms of the synthesis, nucleation, growth, and surface chemistry of CuO NPs is required. Also, new means must be applied to regulate their toxicity, surface chemistry, and biocompatibility for food packaging applications. Future studies in the field of nanoscience and nanotechnology on producing CuO nanofillers should focus on eco-friendly synthesis strategies (renewable and non-toxic resources) that are cost-effective and involve nanomaterials that are easier to fabricate. Further, a scale-up approach must be incorporated to transform laboratory-based research into translational research that benefits mankind and the ecosystem.

The enhanced antimicrobial effects of CuO nanofillers acting against various foodborne contaminants could potentially revolutionize the food packaging sectors. As CuO research and development continues to advance, the fabrication of CuO nanofillers for integration in films/composites is expected to offer substantial growth opportunities in the near future. Indeed, the expansion of metal-doped, polymer-doped, and bio CuO nanofillers being incorporated into FPS in the coming years is expected to bring more industrial applications. More in-depth experimental studies are needed to assess the toxicological risks of CuO-based FPS before their commercialization.

**Author Contributions:** Conceptualization, K.G., G.S. and L.X.; investigation, K.G. and L.X.; writing, G.S. and K.G.; funding, L.X. All authors have read and agreed to the published version of the manuscript.

**Funding:** This research was supported by the innovation platform for Academicians of Hainan Province and Chongqing Engineering Research Center for Micro-Nano Biomedical Materials and Devices.

**Institutional Review Board Statement:** Not applicable.

**Informed Consent Statement:** Not applicable.

**Data Availability Statement:** Data are contained within the article.

**Conflicts of Interest:** The authors declare no conflicts of interest.

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
