# Peer review of "An Overview of the Copper Oxide Nanofillers Integrated in Food Packaging Systems"

_coatings, doi:10.3390/coatings14010081_

Round 1
Reviewer 1 Report
Comments and Suggestions for Authors
Dear Authors
Your manuscript is very nice organized and comprehensive. The topic is popular and such food packaging systems will be explore more and more in the future. More particularly Cu-oxide NPs will shine more as potential nanofillers.
The article consists of important headings/subheadings, and concluding remarks are well presented.
The reference lists is well selected and important reference titles are properly cited in the body text.
Thank you.
Author Response
Comment: Your manuscript is very nice organized and comprehensive. The topic is popular and such food packaging systems will be explore more and more in the future. More particularly Cu-oxide NPs will shine more as potential nanofillers. The article consists of important headings/subheadings, and concluding remarks are well presented. The reference lists is well selected and important reference titles are properly cited in the body text.
Response: We thank the reviewer for his/her clear understanding and consideration of our review article
Reviewer 2 Report
Comments and Suggestions for Authors
The manuscript titles “Copper oxide nanofillers integrated composites/films in food packaging applications: an overview” summarizes the preparation of various CuO nanofillers integrated/coated polymeric food packaging systems with improved mechanical and antimicrobial effects. The topic is relevant and the review deserves consideration. However, at the current state the manuscript needs consideration. The main comments and recommendations are listed below.
The authors briefly described the current state of the topic and justified its’ relevance. However, practical importance of the topic should be showed and proved as well. What is the situation in the market? Who produces films for food packaging modified with bioactive nanofillers? How the consumers react to nanopackaging? For reference: https://doi.org/10.1016/j.fufo.2022.100191, https://doi.org/10.3390/nano11020292
More discussion with direct examples (references) is needed about aggregation and stabilization of CuO NPs. What stabilizers were found better for CuO NPs? What molecular complexes of CuO NPs are not suitable for food packaging?
Section 4.2 “Degradation and ecological impact” is very important for the topic and should be significantly expanded. The authors should discuss with direct examples (references) the materials that were used for food packaging modified with NPs (CuO NPs) and to compare their biodegradation rate, diffusion of NPs to product of distribution in the environment. Potential toxicity effect caused by films biodegradation should be discussed as well.
Figures 1,3,6: the text font should be improved to make the figures readable
The text should be checked by native English speaker for typos and grammatical errors.
Comments on the Quality of English LanguageThe text should be checked by native English speaker for typos and grammatical errors.
Author Response
The manuscript titles “Copper oxide nanofillers integrated composites/films in food packaging applications: an overview” summarizes the preparation of various CuO nanofillers integrated/coated polymeric food packaging systems with improved mechanical and antimicrobial effects. The topic is relevant and the review deserves consideration. However, at the current state the manuscript needs consideration. The main comments and recommendations are listed below.
We thank the reviewer for his valuable comments and suggestions for the comments for the betterment of our article.
Comment 1: The authors briefly described the current state of the topic and justified its’ relevance. However, practical importance of the topic should be showed and proved as well. What is the situation in the market? Who produces films for food packaging modified with bioactive nanofillers? How the consumers react to nanopackaging? For reference: https://doi.org/10.1016/j.fufo.2022.100191, https://doi.org/10.3390/nano11020292
Response 1: As per the reviewer’s comment, we have included the following points in the revised manuscript to address the queries based on suggested references.
Recent reports notified that the increasing demand for FPS has a worldwide market worth more than $300 billion with a 5.2% annual growth rate. Among the FPS, plastic packaging materials dominate the food packaging industry with 61.2% of the global market, alarmingly indicating future environmental and pollution concerns.
Many polymers, inorganic nanomaterials, and biomacromolecules were largely demonstrated for active and smart food packaging applications.
Moreover, adequate promotional activities to raise customer understanding and awareness of nano-packaging will streamline this innovative technology as a sustainable approach.
Comment 2: More discussion with direct examples (references) is needed about aggregation and stabilization of CuO NPs. What stabilizers were found better for CuO NPs? What molecular complexes of CuO NPs are not suitable for food packaging?
Response 2: We believe that the hydrothermal and biogenic fabrication methods were considered effective within the available literature on the synthesis of CuO nanofillers for FPS. However, as mentioned in the article, more research should be demonstrated on optimizing various reaction conditions, complexes of CuO, and stabilizing agents to develop an eco-friendly process.
Comment 3: Section 4.2 “Degradation and ecological impact” is very important for the topic and should be significantly expanded. The authors should discuss with direct examples (references) the materials that were used for food packaging modified with NPs (CuO NPs) and to compare their biodegradation rate, diffusion of NPs to product of distribution in the environment. Potential toxicity effect caused by films biodegradation should be discussed as well.
Response 3: No direct research investigation demonstrates the degradation and recycling process of CuO nanofillers integrated FPS. However, per the reviewer’s suggestion, we have included the following points regarding the future research directions with relevant references.
The biodegradation and recycling process of CuO-nanofiller integrated FPS is in its infant stage. To minimize environmental risks, more in-depth experimental studies should be conducted on integrating CuO nanofillers in biodegradable polymeric FPS and their de-gradable mechanism [94, 95].
Comment 4: Figures 1,3,6: the text font should be improved to make the figures readable
Response 4: All the figure qualities were improved
Comment 5: The text should be checked by a native English speaker for typos and grammatical errors.
Response 5: We thank the reviewer for his/her suggestions. The language of the whole manuscript has been checked with advanced grammar-checking tools.
Reviewer 3 Report
Comments and Suggestions for Authors
The paper is a review that systematizes the data concerning copper oxide nanoparticles (CuPNPs) that are used in food packaging application due to their antibacterial and antimicrobial effects. The review was well structured and approaches the synthesis, characterization, surface modification of diverse CuONPs and fabrication of CuO nanofillers integrated films and composites. Their antibacterial and toxicological properties as well as their adverse side effects and the possibility to combat the situation were reviewed. The CuO-hybrid nanofillers found applications in food packaging industries due to their antibacterial antimicrobial and properties.
The review is of interest for the researchers and specialists in food industry and is in the profile of Coatings journal.
I recommend the following minor corrections:
1)Page 3, line 124
The abbreviation of SPR was not given.
2)Page 4 , lines 147-151
In the following phrase “Doping plays a crucial role in modulating different properties of the NPs due to the following motives: (1) Modification of the electronic structure and surface state, (2) Incorporation of new functional groups, (3) Enhanced catalytic properties, (4) Improved carriers, (5) Greater stability (6) Better biocompatibility [54].”
I recommend the reformulation of “improved carriers”.
3)Page 7, line 250
The abbreviation of LDPE was not given at the first mention.
4)Page 10 , line 381
The abbreviation of PPH was not given.
5)Page 11, line 404
The abbreviation of CMCS was not given.
Author Response
The paper is a review that systematizes the data concerning copper oxide nanoparticles (CuPNPs) that are used in food packaging application due to their antibacterial and antimicrobial effects. The review was well structured and approaches the synthesis, characterization, surface modification of diverse CuONPs and fabrication of CuO nanofillers integrated films and composites. Their antibacterial and toxicological properties as well as their adverse side effects and the possibility to combat the situation were reviewed. The CuO-hybrid nanofillers found applications in food packaging industries due to their antibacterial antimicrobial and properties.
We thank the reviewer for his/her detailed evaluation and report
Comment 1: The review is of interest for the researchers and specialists in food industry and is in the profile of Coatings journal.
I recommend the following minor corrections:
1)Page 3, line 124
The abbreviation of SPR was not given.
2) Page 4 , lines 147-151
In the following phrase “Doping plays a crucial role in modulating different properties of the NPs due to the following motives: (1) Modification of the electronic structure and surface state, (2) Incorporation of new functional groups, (3) Enhanced catalytic properties, (4) Improved carriers, (5) Greater stability (6) Better biocompatibility [54].”
I recommend the reformulation of “improved carriers”.
3)Page 7, line 250
The abbreviation of LDPE was not given at the first mention.
4)Page 10 , line 381
The abbreviation of PPH was not given.
5)Page 11, line 404
The abbreviation of CMCS was not given.
Response 1: As per the reviewer's suggestion, we have compiled all the minor changes and checked the abbreviations throughout the manuscript
Reviewer 4 Report
Comments and Suggestions for Authors
Dear Author,
I suggest a minor revision in relation to the following remarks/suggestions for manuscript coatings-2784608 improvement:
1. I suggest title change to: Cooper oxide nanofillers integrated food packaging systems: An overview
2. Please delete key words "UV blocking" and "water vapor resistance". Add "synthesis" and "antibacterial packaging".
3. Line 39: change "shielding" to "protection"
4. Line: 88-89: you mention a lot of techniques for obtaining CuO NPs and in the following text you describe in great detail hydrothermal method and not the other methods. Why?
5. Generally, for the reader part 2.1. is the most obscure. I suggest that you present part of the data in a table, starting with laser ablation, hydrothermal technique, ... all the way to metal-doped CuO with adequate text that will accompany the table.
6. Subtitle 2.2. Start with "Effects of integration of ..."
7. Line 196-209: Too general part, belongs more to the introduction. Either shorten or modify to match the subtitle on antibacterial FPS.
Figure 4: "water resistance" change to "improved water resistance" and "tensile strength" change to "tensile strength increase"
8. Line 280: Explain the abbreviation TNT
9. Line 341-343: add a reference
10. Line 404: Explain the abbreviation CMCS-SPA-CuO
11. Line 405: explain the abbreviation MTT assay
Kind regards
Author Response
I suggest a minor revision in relation to the following remarks/suggestions for manuscript coatings-2784608 improvement:
Comment 1: I suggest title change to: Cooper oxide nanofillers integrated food packaging systems: An overview
Response 1: As per reviewer suggestion, we have modified the title of this review article.
Comment 2: Please delete key words "UV blocking" and "water vapor resistance". Add "synthesis" and "antibacterial packaging".
Response 2: We thank the reviewer for this suggestion, which has been compiled in the revised manuscript.
Comment 3: Line 39: change "shielding" to "protection"
Response 3: Yes, compiled
Comment 4: Line: 88-89: you mention a lot of techniques for obtaining CuO NPs and in the following text you describe in great detail hydrothermal method and not the other methods. Why?
Response 4: Regarding the CuONPs-based FPS, the available literature studies primarily demonstrated the hydrothermal and biogenic processes. Therefore, we have emphasized these methods more elaborately. Moreover, the studies suggest that uniform size/shape distribution and compatible surface properties are crucial in FPS applications. Therefore, most of the research works focus on the hydrothermal-assisted biogenic process to attain sustainable technologies.
Comment 5: Generally, for the reader part 2.1. is the most obscure. I suggest that you present part of the data in a table, starting with laser ablation, hydrothermal technique, ... all the way to metal-doped CuO with adequate text that will accompany the table.
Response 5: As suggested by the reviewer, this section has been condensed. Furthermore, this review focuses entirely on CuO nanofillers explicitly created for food packaging applications. As a result, we consider that a tabular comparison of data gathered for various synthesis procedures and applications will be unnecessary for the article's objective.
Comment 6: Subtitle 2.2. Start with "Effects of integration of ..."
Response 6: Yes, compiled
Comment 7: Line 196-209: Too general part, belongs more to the introduction. Either shorten or modify to match the subtitle on antibacterial FPS.
Response 7: Yes, compiled
Comment 8: Figure 4: "water resistance" change to "improved water resistance" and "tensile strength" change to "tensile strength increase"
Response 8: Yes, compiled
Comment 9: Line 280: Explain the abbreviation TNT
Response 9: Yes, compiled
Comment 10: Line 341-343: add a reference
Response 10: Yes, compiled
Comment 11: Line 404: Explain the abbreviation CMCS-SPA-CuO
Response 11: Yes, compiled
Comment 12: Line 405: explain the abbreviation MTT assay
Response 12: Yes, compiled
Reviewer 5 Report
Comments and Suggestions for Authors
Please, see the comments in annex.

The quality of the English Language is good.
Author Response
Response: We have compiled all the minor corrections. The FDA approval for CuO NPs is currently under phase 1 and 2 evaluation.
As per reviewers the following points were given in the manuscript
The high stability of CuO nanoparticles is an essential criterion for their uniform size distribution onto composite films and for limiting the Cu migration into food products. It was noticed that the methylcellulose films integrated with gelatin-stabilized CuONPs exhibit only 0.12 µg/mg of Cu migration, which can be further reduced by keeping the food products in a refrigerator at 0-4 °C conditions.
